# An Optimal Algorithm for Marginalization in Bayesian Networks

## Abstract

We study the problem of marginalization in Bayesian networks: given a Bayesian network $G = (V, E)$ and nodes $S$ we wish to marginalize, what is the most compact Bayesian network $G'$ over nodes $V \setminus S$ which is faithful to the independencies and ancestral relationships in the original graph. Efficient solutions to this problem are crucial for the problem of abstraction in Bayesian networks. Prior approaches based on Shachter's topological operations are sensitive to user-chosen node removal and edge reversal orders, provide no optimality guarantee, and can be prohibitively slow when searched exhaustively. We present a novel algorithm for marginalization with the first proof of optimality for algorithms of its kind, with an empirical speedup of up to several orders of magnitude over the prior state-of-the-art.

## 1 Introduction

Bayesian networks are a class of probabilistic graphical models which use directed acyclic graphs (DAGs) to represent multivariate distributions with highly structured interdependencies. In a Bayesian network, the nodes of a graph are random variables, and the edges of the graph represent conditional probability distributions. Bayesian networks provide a natural framework for encoding the conditional independencies found in a joint probability distribution, allowing complex distributions to be represented more compactly (Koller & Friedman, 2009; Russell & Norvig, 2021).

Bayesian networks are a class of probabilistic graphical models which use directed acyclic graphs (DAGs) to represent multivariate distributions with highly structured interdependencies. In a Bayesian network, the nodes of a graph are random variables, and the edges of the graph represent conditional probability distributions. Bayesian networks provide a natural framework for encoding the conditional independencies found in a joint probability distribution, allowing complex distributions to be represented more compactly (Koller & Friedman, 2009; Russell & Norvig, 2021).

While much work is being done on improving the interpretability of transformer-based models that dominate today's large language models, their internal representations remain largely opaque (Zhao et al., 2024). In contrast, Bayesian networks offer, at least in principle, the potential for more interpretable representations. Each node corresponds to a specific random variable that can be directly analyzed, and it is possible for users to trace how observed evidence propagates through the network and influences the posterior distribution. However, one challenge that Bayesian networks face is that both exact and approximate probabilistic inference in Bayesian networks are NP-hard (Cooper, 1990; Dagum & Luby, 1993).

One way to tackle this challenge is to perform abstraction in Bayesian networks, where a large, intractable Bayesian network is simplified into a smaller one in which inference is tractable. More specifically, one method of abstraction is *marginalization*, the practice of removing or 'marginalizing out' a subset of the nodes of the original network (Yet & Marsh, 2014). Marginalization can also be useful in circumstances where the value of certain random variables are unable to be observed or irrelevant to the task at hand.

There are several factors to consider when marginalizing a Bayesian network. Firstly, the original network encodes a particular set of conditional independencies, and so our marginalized Bayesian network must not encode any independencies that are not present in the original network. If we do encode spurious independencies, our model will no longer be faithful and may be unable to correctly represent the true marginal distribution over the remaining random variables.

On the other hand, we also wish to have as compact a graph as possible. A trivial solution to the problem of finding a graph that does not encode any spurious conditional independencies is a complete graph, where there is an edge between every two vertices, and so there are zero encoded conditional independencies. However, each additional edge in the marginalized graph requires storing larger and more complex conditional probability distributions, resulting in a less compact representation.

Finally, in practice, Bayesian networks do not only encode a particular set of conditional independencies, but are also often designed to reflect an intuitive generative or causal process underlying the distribution over the variables (Pearl, 2009). So another desirable property we may seek in our marginalized graph is for the topological ordering implied by the base graph to be preserved.

For this work, we focus on minimizing the number of edges as a proxy for graph compactness, though our approach can be generalized to other measures of graph complexity. Our problem can then be stated as follows: given a Bayesian network $G = (V, E)$ and a set of nodes $S$ that we wish to marginalize, what is the Bayesian network $G'$ on the nodes $V \setminus S$ such that $G'$ does not encode any conditional independencies that $G$ does not encode, $G'$ does not contradict the ancestral relationships established in $G$, and $G'$ has the minimum number of edges?

The prior state-of-the-art algorithm for this problem is based on Shachter's topological operations introduced in Shachter (1986) and later applied to Bayesian network abstraction by Yet & Marsh (2014). There are two main limitations of this work. First, the algorithm leaves certain runtime choices up to the user, particularly node removal order and edge reversal order (which we discuss later), with the choices impacting the compactness of the marginalized graph. As a result, to find the optimal graph, the user must conduct a brute-force search over these runtime choices. Secondly, there is no proof that the algorithm finds the optimal marginalized graph with minimal edges, even with a brute-force search over the runtime choices.

In response, we present a novel algorithm for Bayesian network marginalization, with, to our knowledge, the first proof of optimality for such an algorithm. Our method is guaranteed to produce a marginalized graph with the minimal number of edges while faithfully preserving the independencies and ancestral relationships of the original graph. Furthermore, by conducting a more structured search over the space of marginalized graphs, our algorithm empirically attains a substantial speedup over the prior state-of-the-art across a wide class of Bayesian networks, with speedups in some cases reaching multiple orders of magnitude.

## 2 BACKGROUND

### 2.1 DEFINITIONS

A *Bayesian network* is comprised of nodes representing random variables and directed edges representing conditional probabilities, with the restriction that these directed edges cannot form a cycle. If we have $A \to B$, then we say that $A$ is a *parent* of $B$ and $B$ is a *child* of $A$, denoted $A \in Pa(B)$ and $B \in Ch(A)$, respectively. If we have $A \to \cdots \to B$, then we say that $A$ is an *ancestor* of $B$ and $B$ is a *descendant* of $A$, denoted $A \in an(B)$ and $B \in de(A)$, respectively. Note that this notation can be generalized, so $an(\{B, C\})$ denotes the union of $an(B)$ and $an(C)$.

A *path* between two vertices $A$ and $B$ is a sequence of vertices $\pi = (A = A_1, A_2, \ldots, A_{n-1}, A_n = B)$ such that $A_i$ and $A_{i+1}$ are connected by an edge for all $i$. Given any intermediate node $A_i$ on this path, we call $A_i$ a *collider* if we have $A_{i-1} \to A_i \leftarrow A_{i+1}$, and a *non-collider* otherwise.

A path $\pi$ is *active* given a set $Z$ if all of the non-collider nodes on $\pi$ are not in $Z$, and all of the collider nodes on $\pi$ are in $an(Z)$. We say that $A$ is independent of $B$ given a set $Z$, denoted $A \perp B \mid Z$, if it is true that there exists no active path from $A$ to $B$ given $Z$. This is a *conditional independence*.

In addition to Bayesian networks, another category of probabilistic graphical models is *ancestral graphs* (Richardson & Spirtes, 2002). In addition to directed edges, these are allowed to contain bidirected edges $\leftrightarrow$. Ancestral graphs must be acyclic and if we have $A \leftrightarrow B$, then we cannot have $A \in an(B)$ or $B \in an(A)$. We will call two nodes *adjacent* if they are connected by an edge of any type.

A specific subclass of ancestral graphs are maximal ancestral graphs, or *MAGs*. Every ancestral graph can be turned into an equivalent MAG (Ali & Richardson, 2002).

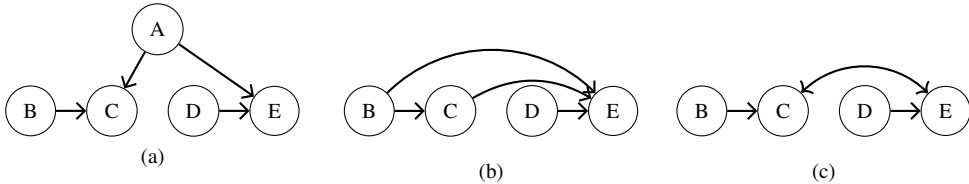

Figure 1: (a) is a starting Bayesian network. If we marginalize out node $A$, a marginalized Bayesian network is in (b), but this has strictly less conditional independencies than the MAG in (c).

MAGs have the additional property that if $A$ and $B$ are not adjacent, then there must exist some set $Z$ such that $A \perp B \mid Z$. Paths have the same definition for MAGs as Bayesian networks—there must exist an edge between adjacent nodes in the path. For MAGs, we call $A_i$ a collider if we have $(A_{i-1} \to A_i$ or $A_{i-1} \leftrightarrow A_i)$ and $(A_i \leftarrow A_{i+1}$ or $A_i \leftrightarrow A_{i+1})$. For convenience, we will denote this as having $A_{i-1} \overset{?}{\to} A_i \overset{?}{\leftarrow} A_{i+1}$, where the question marks denotes that the statement holds regardless of what the edge endings at $A_{i-1}$ and $A_{i+1}$ are. As before, all nodes that are not colliders are non-colliders. Our definition for an active path remains the same, as does the definition for conditional independence.

A *topological ordering* is an ordering of the nodes in an ancestral graph (or Bayesian network) such that if $A \to B$, then $A$ appears before $B$ in the ordering. Alternatively, this is phrased as $A$ having a lower topological order than $B$. It is possible for an ancestral graph to have more than one possible topological ordering, and each ancestral graph must have at least one topological ordering.

We wish for our marginalized Bayesian network to not contradict the ancestral relationships established in the original relationship. Essentially, this means that if $A \in an(B)$ in $G$, we cannot have $B \in an(A)$ in $G'$, as this would reverse the direction of causality in our network.

## 2.2 MARGINALIZATION

One challenge in finding faithful marginalized Bayesian networks with the minimal number of edges is that Bayesian networks are not closed under marginalization (Koster, 2002). In other words, for some choices of Bayesian network $G = (V, E)$ and nodes to remove $S$, it is impossible to find a $G'$ such that the conditional independencies of $G'$ are the same as the conditional independencies of $G$ amongst the nodes in $V \setminus S$. A classic example of this phenomena occurs with the Bayesian network shown in Figure 1(a) and marginalization set $\{A\}$ (Richardson & Spirtes, 2002). It can be shown that all possible marginalized graphs are missing at least one conditional independency that the original graph had; this is proven in Lemma A.2. One such example of a marginalized graph is shown in Figure 1(b). In this $G'$, we have $B \not\perp E$, but in our original graph, $B$ and $E$ are independent.

On the other hand, MAGs are closed under marginalization (Ali & Richardson, 2002). Continuing our example, Figure 1 is the $G_{MAG}$ that encodes the same conditional independencies as $G$ amongst nodes $B, C, D, E$. This is because any active path in $G$ that included node $A$ can still be taken in $G_{MAG}$, as the bidirected edge allows us to have an arrowhead at $C$ and at $E$.

Because every Bayesian network is also a MAG, we can find a MAG on the nodes $V \setminus S$ that encodes exactly the conditional independencies as those encoded by $G$ amongst the vertices in $V \setminus S$. This provides the motivation for our method: we first process the graph and obtain an equivalent MAG after removing our nodes, and then we find our Bayesian network by simplifying from this MAG.

## 3 RELATED WORK

### 3.1 ABSTRACTION WITHIN BAYESIAN NETWORKS

Four abstraction methods for Bayesian networks have been identified in earlier work (Yet & Marsh, 2014): node removal, node merging, state-space collapsing, and edge removal. The latter two methods involve consultation with experts, and are thus impractical when such resources are unavailable.

In node merging, two nodes $A$ and $B$ are combined to form a merged random variable $AB$. However, the state space of $AB$ is simply the Cartesian product of the state spaces of $A$ and $B$. Therefore, although this reduces the number of nodes, it does not allow for a more compact representation of the probability distribution, as we now must store the joint distribution of $A$ and $B$.

Node removal is based on Shachter's topological operations (Shachter, 1986; 1988). Shachter proved that the following operations do not introduce new conditional independencies to a graph:

1. Removing barren nodes, or nodes that do not have any children.

2. Reversing covered edges. An edge $A \rightarrow B$ is covered if $Pa(A) \cup \{A\} = Pa(B)$; we are allowed to replace $A \rightarrow B$ with $A \leftarrow B$ for such edges.

3. Adding edges.

Using this, one method to remove a node $R$ is that, for each of its children $C_i$, add edges from the parents of $R$ to $C_i$ and the parents of $C_i$ to $R$, and then replace $R \rightarrow C_i$ with $R \leftarrow C_i$. After doing this for all of $R$'s children, it will be barren, and we can remove $R$.

Therefore, given a set $S$ of nodes to remove, the process of node removal above provides a valid method of marginalizing the graph that results in a $G'$ without false conditional independencies. However, the biggest shortcoming of using this method to perform marginalization is that it does not guarantee that we will be able to obtain the optimal (most compact) marginalized Bayesian network, even if we test all possible orders of reversing edges and removing nodes.

### 3.2 Other probabilistic graphical models

Ancestral graphs were developed with the motivation to generalize Bayesian networks and allow for faithful representations of marginalized Bayesian networks or Bayesian networks with latent variables (Richardson & Spirtes, 2002). Other models have also been developed for this purpose (Koster, 2002; Richardson, 2003), but ancestral graphs, and specifically, MAGs, have received the most attention in the literature (Richardson & Spirtes, 2002; Ali & Richardson, 2002).

A more generalized version of MAGs are *acyclic directed mixed graphs,* or ADMGs (Richardson, 2003). These must still be acyclic, but they are allowed to have multiple edge types between two nodes, such as $A \rightarrow B$ and $A \leftrightarrow B$. These are also closed under marginalization. An advantage of ADMGs is that they are more computationally efficient to obtain from a Bayesian network with nodes that we wish to marginalize (Koster, 2002), but the structure of MAGs is more similar to that of Bayesian networks, and thus easier to reduce from.

An alternative to Bayesian networks are undirected graphs, which only have undirected edges between the nodes. These are closed under marginalization (Richardson & Spirtes, 2002). However, the drawback of undirected graphs is that they cannot represent the direction of causality, and can thus only capture correlations.

## 4 Optimal marginalization in Bayesian networks

We now introduce an optimal algorithm for marginalization in Bayesian networks. Specifically, our algorithm solves the following problem: Given a Bayesian network $G = (V, E)$ and a set $S$ of nodes to remove, return a set of Bayesian networks $G_1, \ldots, G_n$ satisfying the following properties:

 (i) Each $G_i$ has nodes $V \setminus S$.

 (ii) The conditional independencies encoded by $G_i$ are a subset of the conditional independencies encoded by $G$.

 (iii) The ancestral relationships established in $G$ are preserved.

 (iv) There does not exist any Bayesian network $G'$ satisfying the aforementioned three points which has less edges than $G_i$.

First, we will provide an overview for our algorithm given in Algorithm 1. We know that MAGs are closed under marginalization, so there exists a marginalized graph in the form of a MAG that has the

---

**Algorithm 1** Marginalization via MAGs

---

    **Input:** A Bayesian network $G = (V, E)$, a set of nodes to remove $S \subseteq V$.
    **Output:** A set of minimal marginalized graphs *final_graphs*.
1: **function** FINDMINIMALGRAPHS($G, S$)
2:     $G_{MAG} \leftarrow$ MARGINALIZETOMAG($G, S$)                 ▷ via Hu Evans
3:     $\mathcal{C} \leftarrow$ FINDBIDIRECTEDCOMPONENTS($G_{MAG}$)       ▷ Sets of nodes connected by $\leftrightarrow$ paths
4:     $G_{base} \leftarrow G_{MAG}$ with all bidirected ($\leftrightarrow$) edges removed.
5:     *component_solutions* $\leftarrow$ []
6:     **for** each component $C \in \mathcal{C}$ **do**
7:         *solutions_by_cost*$_C \leftarrow$ dict()          ▷ keys are num. edges added, values are list of graphs
8:         $\Pi \leftarrow$ ALLTOPOLOGICALORDERINGS($C$)
9:         **for** each ordering $\sigma = (X_1, \ldots, X_n)$ in $\Pi$ **do**
10:             $G_{C,\sigma}, k_{C,\sigma} \leftarrow$ GETEDGESFORORDERING($C, \sigma$)         ▷ See algorithm 2
11:             Append $G_{C,\sigma}$ to *solutions_by_cost*$_C$.get($k_{C,\sigma}$, [])
12:         Append *solutions_by_cost*$_C$ to *component_solutions*
13:     *final_graphs* $\leftarrow$ []
14:     *min_cost* $\leftarrow \infty$
15:     *cost_combinations* $\leftarrow$ COSTCOMBINATIONS(*solutions_by_cost*)         ▷ Uses a heap
16:     **for** each cost tuple $\rho = (c_1, \ldots, c_k)$ in *cost_combinations* **do**
17:         *current_cost* $\leftarrow \sum_{i=1}^{k} c_i$
18:         **if** *current_cost* $>$ *min_cost* **then**
19:             **break**
20:         **for** $(G_{C_1}, \ldots, G_{C_k})$, where $G_{C_i}$ has cost $c_i$ for all $i$ **do**
21:             $G_{final} \leftarrow G_{base} \cup \bigcup_{i=1}^{k} G_{C_i}$
22:             **if not** $G_{final}$ has cycles **then**
23:                 Append $G_{final}$ to *final_graphs*
24:                 *min_cost* $\leftarrow$ *current_cost*
25:     **return** *final_graphs*

---

exact conditional independencies of our original graph on $V \setminus S$. Therefore, we will first convert our Bayesian network to a MAG using an existing algorithm (Hu & Evans, 2020).

The resulting graph $G_{MAG}$ will have directed and bidirected edges. Let the *base graph* contain only the directed edges of $G_{MAG}$. We know that our final graph $G'$ must contain all edges in the base graph, or else $G'$ will have a conditional independency that $G_{MAG}$ did not have (cf. Lemma A.1). Therefore, we just need to focus on converting the bidirected edges in $G_{MAG}$ into directed edges.

The crucial structures to identify in $G_{MAG}$ are *bidirected connected components*, or BCCs. These are subsets of the nodes of $G_{MAG}$, such that $A$ and $B$ are in the same BCC if and only if there exists a path from $A$ to $B$ consisting only of bidirected edges.

The crux of our algorithm is determining the ways in which each BCC can be reduced. This can be thought of as converting the bidirected edges to directed edges (and adding additional directed edges, as necessary). If our BCC only had bidirected edges, there would be no inherent topological order amongst nodes in the same BCC. On the other hands, any directed edges present amongst nodes in a BCC constrain the set of valid topological orderings over the variables. For each BCC, our algorithm then searches over all valid topological orderings and computes the directed edges that must be added in order to have a reduction compatible with that topological ordering.

After performing this for each component, we'll have a list of valid reductions for each component. Due to the disjoint manner of the components, the edges added for different components will also be disjoint. Therefore, given a choice of reduction for each component, the number of edges in our final graph $G'$ is just a sum of the the number of edges in each component's reduction, as well as the number of edges in the base graph.

After filtering out the final graphs that have the minimum number of edges, we will test for cycles, and discard the graph if a cycle exists. If necessary, we will continue searching amongst the reductions that result in the second lowest number of edges in the final graph, and so on, stopping once we can return a non-empty collection of acyclic Bayesian networks.

---

**Algorithm 2** Generate Edges for a Given Topological Ordering

---

**Input:** A bidirected component $C$ and a topological ordering $\sigma = (X_1, \ldots, X_n)$ of $C$.
**Output:** A faithful Bayesian network for the component $G_C$, the number of added edges $k$.
1: **function** GETEDGESFORORDERING($C, \sigma$)
2:    $G_C \leftarrow$ Subgraph of $G_{MAG}$ induced by $C \cup \text{Parents}(C)$ initialized with no edges
3:    $m \leftarrow 0$                                                    ▷ Counter for number of edges
4:    **for** $i \leftarrow 1$ to $n$ in decreasing order **do**
5:      **for** $j \leftarrow 1$ to $i - 1$ **do**
6:        **if** exists bidirected path from $X_i$ to $X_j$ along nodes w/ lower top. order than $X_i$ **then**
7:          Add edge $X_j \rightarrow X_i$ to $G_C$ and update $m \leftarrow m + 1$ if $X_j \rightarrow X_i$ doesn't exist
8:          Add edge $V \rightarrow X_i$ to $G_C$ and update $m \leftarrow m + 1$ for all $V \in Pa(X_j)$ if $\not\exists V \rightarrow X_i$
9:    **return** $G_C, m$

---

In Algorithm 2, we give our procedure for determining which edges must be added given a component and a topological ordering for that component. For any component $C$, it is sufficient to add edges only between nodes in $C \cup Pa(C)$.

Given a topological order $\sigma = (X_1, \ldots, X_n)$ on the nodes of the BCC, we process the nodes in reverse, adding edges with an arrowhead at each node for each step. In other words, we add edges to node $X_i$, first for $i = n$, then $i = n - 1$, and so on. Given any other node $X_j$ with $j < i$, we add edge $X_j \rightarrow X_i$ if and only if there exists some bidirected path $X_j \leftrightarrow X_{j_2} \leftrightarrow \cdots \leftrightarrow X_{j_{l-1}} \leftrightarrow X_i$ such that $X_{j_2}, \ldots, X_{j_{l-1}}$ all have lower topological order than $X_i$. In our algorithm, this is equivalent to there being no node on the path such that $X_j \rightarrow X_{j_l} \leftarrow X_i$.

Lastly, if we must add $X_j \rightarrow X_i$, our algorithm also adds edges from each of $X_j$'s parents to $X_i$.

## 4.1 PROOF OF CORRECTNESS

First, we show that the graphs returned in *final_graphs* are valid marginalized Bayesian networks.

**Theorem 4.1.** *Take a Bayesian network $G = (V, E)$ and a set of nodes to remove $S \subseteq V$, and let* final_graphs *be the output of running Algorithm 1 on $G, S$. For each $G'$ in* final_graphs*, $G'$ is a Bayesian network, and there exists no conditional independency encoded by $G'$ not encoded by $G$.*

We include a quick proof sketch below. The complete proof is included in the appendix.

**Proof Sketch.** First, we need to show that $G'$ is a Bayesian network. Since neither the base graph nor any of the component graphs contain any bidirected edges, $G'$ must only contain directed edges. Additionally, it must be a DAG because of the check we perform before returning *final_graphs*.

In order to show that $G'$ encodes no extra conditional independencies, it suffices to show that if there exists an active path between $X_i$ and $X_j$ given $Z$ in $G$, then there exists such an active path in $G'$. Moreover, since $G$ and $G_{MAG}$ have the same active paths, it suffices to show that we can find corresponding active paths in $G'$ for all active paths in $G_{MAG}$.

Because $G'$ has the same directed edges as $G_{MAG}$, it suffices to focus on segments of the path that intersect with a BCC. For any active path $\pi = (X_j, X_{j_2}, \ldots, X_{j_{m-1}}, X_i)$ from $X_j$ to $X_i$ in $G_{MAG}$, we are able to find an active path $\pi'$ in $G'$ because of how we've added edges in our algorithm; we must either have $X_j \rightarrow X_i$ or $X_j \rightarrow X_{j_l} \leftarrow X_i$ for some $X_{j_l}$ on $\pi$ (see Lemma A.3). Since we can always find a corresponding active path, there cannot be new conditional independencies in $G'$. □

## 4.2 PROOF OF OPTIMALITY

Now that we've shown that all of the graphs we return in *final_graphs* are valid marginalized graphs, we wish to show that our algorithm will return the optimal marginalized Bayesian network.

**Theorem 4.2.** *Take a Bayesian network $G = (V, E)$ and a set of nodes to remove $S \subseteq V$, and let* final_graphs *be the output of running Algorithm 1 on $G, S$. For any $G^*$ that satisfies the following conditions, we must have $G^*$ in* final_graphs*:*

- *$G^*$ is a marginalized Bayesian network on the nodes $V \setminus S$.*

- *The conditional independencies encoded by $G^*$ are a subset of the conditional independencies encoded by $G$.*

- *The ancestral relationships established in $G$ are preserved.*

- *$G^*$ contains the minimum number of edges amongst all Bayesian networks satisfying the previous 3 conditions.*

Again, we include a quick proof sketch below. The complete proof is included in the appendix.

**Proof Sketch.** First, we will show that $G^*$ is equal to $G_{base} \cup \bigcup_{i=1}^{k} G_{C_i, \sigma_i}$ for some choices of topological orderings $\sigma_i$ for the components $C_i$. We use Lemma A.1 to argue that $G^*$ must contain all of the edges in $G_{base}$.

Then, since $G^*$ is a Bayesian network, it has a topological ordering, and this ordering will induce a topological ordering amongst the nodes in each of the BCCs of $G_{MAG}$. We will show that given a topological ordering for each component, we must add all of the edges added by our algorithm for any valid $G'$ (and thus those edges must exist in $G^*$), or else we can identify a conditional independency that exists in that $G'$ but not $G_{MAG}$. Because of this, $G^*$ must have all of the edges in $G_{base} \cup \bigcup_{i=1}^{k} G_{C_i, \sigma_i}$, and by the minimality of $G^*$ we know that $G^*$ is equal to $G_{base} \cup \bigcup_{i=1}^{k} G_{C_i, \sigma}$ for some $\sigma$.

Lastly, we argue that our algorithm is able to identify the correct $\sigma$. To do this, we show that we will never add the same edge when processing two different components. Because our algorithm checks if an edge exists in the base graph when processing each component, we know that all of the edges in $\{G_{base}\} \cup \{G_{C_i, \sigma_i}\}_{i=1}^{k}$ are disjoint.

In order to identify the correct $\sigma$, we must find the graph that minimizes the number of edges added to $G_{base}$. In our algorithm, we minimize the sum of the number of edges added to each component graph. However, due to the disjointness of these edges, this is equal to the number of edges added to $G_{base}$. Therefore, we are indeed able to identify and return the desired graph in *final_graphs*. $\square$

## 5 EXPERIMENTS

When compared to the previous method of abstraction described in Section 3.1, our algorithm has the advantage of having a guarantee that we are able to output all of the optimal graphs. In this section, we empirically compare the wall clock time of our algorithm with their algorithm.[1]

Before discussion of the results, let's describe the algorithm based on the previous method. Recall that in that algorithm, we have the freedom to remove nodes in any order, and for each node that we remove, we have the freedom to reverse the edges between it and its children in any order. One useful heuristic that was previously suggesting for choosing which edges to reverse was to do so in an order that minimized the number of edges added (Yet & Marsh, 2014). However, greedily choosing to reverse edges according to this heuristic does not always result in the minimal graph; see section A.3.

Moreover, there is also no proof that the order of node removal would not make a difference. On the other hand, it is known that there exists optimal node elimination orders for Variable Elimination, a method that performs exact inference on Bayesian networks; also, finding such an optimal order is NP-hard, so there is no simple heuristic (Cooper, 1990).

Therefore, in our algorithm for this method, we test all possible edge reversal and node removal orders. To compare the previous method with our method, we ran simulations where we randomly generated our initial Bayesian networks (where edges had a specifiable probability for being added), and randomly selected a subset of nodes to remove.

We had three parameters that we could change: the number of nodes in the initial graph, the probability that an edge was added, and the number of nodes that we were removing. For each of these parameters, we would vary it while keeping the values of the other two parameters fixed at default values. These default values were 10 nodes in the initial graph, an edge probability of 0.4, and 4 nodes removed. For each choice of parameters, we randomly generated 30 graphs, and compared the wall clock time of our method and the previous method.

---

[1]Our experiment code is available at `https://doi.org/10.5281/zenodo.17197254`.

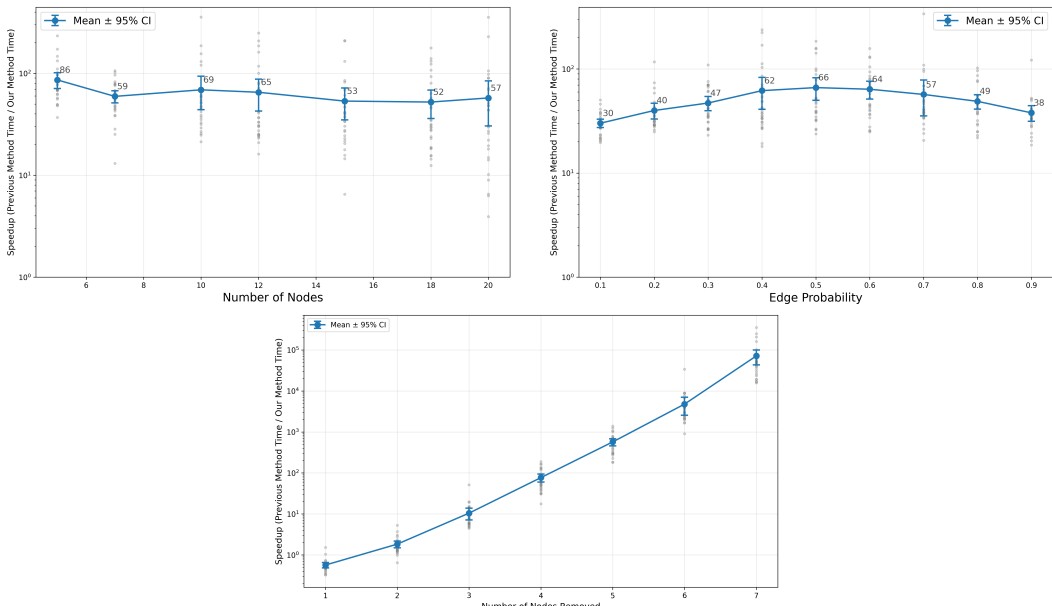

Figure 2: Wall-clock speedup of our proposed algorithm over the prior method as a function of (i) the number of nodes $|V|$ in the original graph, (ii) the edge-inclusion probability $p$ used to generate the original Bayesian network, and (iii) the number of removed nodes $|S|$.

In our plots, we've graphed the *speedup* that our method provides, or the ratio of the time of the previous method to that of our method. The average is plotted and labeled, the 95% confidence interval is annotated, and the gray points allow outliers to be visible.

To summarize the results, our method provides a speedup in all cases except that of one node being removed. In many cases, our method is many orders of magnitude faster than the existing method.

### 5.1 LIMITATIONS AND FUTURE WORK

In the future, we hope to test if the trends demonstrated above still hold up when we increase the number of nodes in our original graph; we were restricted to studying smaller graphs for this paper due to time constraints. Additionally, although our method is substantially faster than the previous method, it can still be cumbersome for very large graphs, and we have not shown that the runtime is sub-exponential. Despite this, marginalization can still be helpful in instances when we wish to ask many inference queries for a marginalized graph. Although there is an upfront cost of creating the marginalized graph, we can reuse the marginalized graph for many rapid, subsequent calculations. For example, we may have a large Bayesian network for stocks, marginalize it to create a smaller network, and run it for predictions under many different circumstances.

Further work is also needed to assess whether converting to a MAG is necessary, or if a similar algorithm can be used to reduce to a Bayesian network from a (non-maximal) ancestral graph or an ADMG, for which the cost of marginalization from a Bayesian network might be cheaper.

## 6 CONCLUSION

In this work, we proposed a solution for the problem of optimal marginalization in Bayesian networks. This is the first algorithm that has a proof for optimality. Prior to this, the best method for marginalization in Bayesian networks was based on Shachter's topological operations. However, there is no optimality guarantee for this method, and due to its reliance on exhaustive search, it can become computationally intensive.

Instead, our method takes advantage of the closure of MAGs under marginalization. We use this to find a MAG that has the exact conditional independencies from our original graph that we wish to encode. The structure of the MAG allows us to have a more systematic method of searching for the optimal graph. We are able to decompose our problem into the subproblems of reducing the bidirected connected components of the MAG.

Additionally, the structure of the MAG allows us to present the first proof of optimality for an algorithm of this purpose. Moreover, empirical results verify the advantages of our approach, exhibiting speedups of up to several orders of magnitude over the baseline.

## 7    Ethics statement

Our work is theoretical, and therefore did not involve human subjects or harmful methodologies. Additionally, our work does not involve any data set releases. Our algorithm is intended to advance fundamental understanding in the field of Bayesian networks, and we are not aware of any immediate discrimination/bias/fairness concerns, privacy issues, or potentially harmful insights/applications.

## 8    Reproducibility statement

As our algorithm is novel, we provide downloadable source code, both as supplementary material, and as a footnote in section 5, including implementations of both our algorithm and the prior state-of-the-art baseline. Because our paper also includes theoretical results, we have included thorough proofs in the appendix below; see sections A.1 and A.2.

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

## A APPENDIX

**Lemma A.1.** *Given a Bayesian network with two nodes $X_i$ and $X_j$ such that $X_i \notin an(X_j)$ and $X_i$ and $X_j$ are not adjacent. If we have any subset of the nodes $Z$ such that $Pa(X_i) \subseteq Z$ and the topological order of each node in $Z$ is lower than that of $X_i$, then*

$$X_i \perp X_j \mid Z.$$

*Proof.* Consider any path from $X_j$ to $X_i$. We wish to show that this path is blocked.

Since $X_j$ and $X_i$ are not neighbors, any such path must contain at least one intermediate node. Consider the node along the path that immediately precedes $X_i$. If this node, say $W$, is a parent of $X_i$, then we have $W \to X_i$ and $W$ must be a non-collider along this path. Thus, the path is blocked at $W$, as $W \in Pa(X_i) \subseteq Z$.

Let's say the node adjacent to $X_i$ along the path is instead a child, so we have $\cdots \leftarrow X_i$. Since we cannot have $X_j \leftarrow \cdots \leftarrow X_i$ due to the constraint that $X_i \notin an(X_j)$, we must have a collider on this path.

Consider the first collider that we encounter as we traverse from $X_i$ to $X_j$, say $C$. We will have $\to C \leftarrow \cdots \leftarrow X_i$, and so $C$ has a higher topological order than $X_i$.

Since all of the elements of $Z$ have a lower topological order than $X_i$, it is impossible for $C$ to be an ancestor of any of the elements of $Z$. Thus, $C \notin an(Z)$, and so the path is blocked at $C$.

Since all possible paths from $X_i$ to $X_j$ are blocked, we indeed have $X_i \perp X_j \mid Z$. $\qquad\square$

**Corollary A.1.1.** *Given a Bayesian network with two nodes $X_i$ and $X_j$ such that $X_i$ and $X_j$ are not adjacent, there must exist some conditioning set $Z$ such that $X_i \perp X_j \mid Z$.*

*Proof.* We know that we cannot have $X_i \in an(X_j)$ and $X_j \in an(X_i)$. Either the first is false, in which case we have $X_i \perp X_j \mid Pa(X_i)$, or the second is false, in which case we have $X_i \perp X_j \mid Pa(X_j)$. $\qquad\square$

**Lemma A.2.** *Consider the graph $G$ depicted in Figure 1(a). After marginalizing node $A$, there exists no Bayesian network $G'$ on nodes $B, C, D,$ and $E$ that has the same conditional independencies as $G$ on nodes $B, C, D,$ and $E$.*

*Proof.* First, we claim that $G'$ must also have $B \to C$ and $D \to E$.

If we do not also have $B \to C$ and $D \to E$ in $G'$, then there will exist $Z_1$ and $Z_2$ such that $B \perp C \mid Z_1$ and $D \perp E \mid Z_2$ by Corollary A.1.1. However, since $B \to C$ and $D \to E$ in $G$, these are conditional independencies that do not hold in $G$. We cannot have conditional independencies in $G'$ but not $G$, so we must have $B \to C$ and $D \to E$.

Secondly, we claim that $G'$ must also have either $C \to E$ or $E \to C$. If $E$ and $C$ were not adjacent in $G'$, Corollary A.1.1 implies that there exists $Z_3 \subseteq G'$ such that $E \perp C \mid Z_3$. However, for any $Z_3$ that does not include $A$, we have $E \not\perp C \mid Z_3$ in $G$, as $C \leftarrow A \to E$ is an active path. Therefore, we know that $C$ and $E$ must be adjacent.

**Case 1: we have $C \to E$.** In this case, note that we cannot have $B \leftarrow E$ in $G'$, as this would introduce cycle $B \to C \to E \to B$. Thus, the only two possibilities are $B \to E$ or $B$ and $E$ not being adjacent. We claim that the latter is impossible. Assume for the sake of contradiction that it were true. Then, we would have $B \perp E \mid C, D$ in $G'$ by Lemma A.1. However, this is not true in $G$, as $B \to C \leftarrow A \to E$ is an active path. Therefore, we must have $B \to E$ in $G'$.

Because $B$ and $E$ are adjacent in $G'$, we know that $B \not\perp E$ in $G'$. However, $B \perp E$ in $G$. Thus, $G'$ has strictly fewer conditional independencies than $G$.

**Case 2: we have $C \leftarrow E$.** Note that we cannot have $C \to D$ in $G'$, as then we would have cycle $C \to D \to E \to C$. Therefore, we must either have $C \leftarrow D$ or $C, D$ cannot be adjacent. We claim that the latter is impossible. Assume for the sake of contradiction that it were true. Then, by Lemma A.1, we know that $C \perp D \mid B, E$. However, this is not true in $G'$, as the path $C \leftarrow A \to E \leftarrow D$ is active. Therefore, we must have $C \leftarrow D$ in $G'$.

But, since $C$ and $D$ are adjacent in $G'$, we know that $C \not\perp D$ in $G'$. Yet, we have $C \perp D$ in $G$. Thus, $G'$ again has strictly fewer conditional independencies than $G$. $\square$

**Lemma A.3.** *Take nodes $X_i$ and $X_j$ in the same BCC in $G_{MAG}$, such that in our topological ordering, $\sigma$, the topological order of $X_i$ is higher than that of $X_j$. In Algorithm 2, we will add edge $X_j \to X_i$ if and only if there exists a bidirected path from $X_j$ to $X_i$ in $G_{MAG}$ with no intermediate node on that path, $X_k$, having edges $X_i \to X_k$ and $X_k \leftarrow X_j$ in $G_C$.*

*Proof.* In Algorithm 2, we will add edge $X_j \to X_i$ if and only if there exists a bidirected path from $X_j$ to $X_i$ in $G_{MAG}$ such that all of the intermediate nodes in this path have lower topological order than $X_j$. Thus, it suffices to show that the following two statements are equivalent:

1. There exists a bidirected path from $X_j$ to $X_i$ such that, in $\sigma$, all of the intermediate nodes in this path have lower topological order than $X_i$.

2. There exists a bidirected path from $X_j$ to $X_i$ in $G_{MAG}$ with no intermediate node $X_k$, having edges $X_i \to X_k \leftarrow X_j$ in $G_C$.

This is equivalent to showing that the logical opposites of these statements are equivalent:

1. For every bidirected path between $X_j$ and $X_i$ in $G_{MAG}$, there exists some intermediate node $X_k$ on the path such that $\sigma(X_k) > \sigma(X_i)$.

2. For every bidirected path in $G_{MAG}$, there exists some intermediate node $X_k$ such that $X_j \to X_k \leftarrow X_i$ in $G_C$.

First, we will show that $1 \implies 2$. Take any arbitrary bidirected path $\pi$ between $X_j$ and $X_i$, and take the node on $\pi$ with maximal topological order. Call this node $X_k$. Note that by the assumption of statement 1, $\sigma(X_k) > \sigma(X_i)$.

In Algorithm 2, we would have processed node $X_k$ before $X_i$, because it has a higher topological order, and we process nodes in reverse topological order. Then, when processing $X_k$, we would have added $X_k \leftarrow X_i$ to $G_C$. This is because along the subset of $\pi_{k:i}$ between $X_k$ and $X_i$, all of the intermediate nodes have lower topological order than $X_k$, by our assumption that $\sigma(X_k)$ was maximal amongst the nodes in $\pi \supset \pi_{k:i}$.

Similarly, note that $\sigma(X_j) < \sigma(X_i) < \sigma(X_k)$, and so all of the intermediate nodes on the path $\pi_{j:k} \subset \pi$ have a lower topological order than $X_k$. Therefore, we will also add edge $X_j \to X_k$ in $G_C$.

Thus, we have found intermediate node $X_k$ such that $X_j \to X_k \leftarrow X_i$ in $G_C$ for $\pi$. Since our choice of $\pi$ was arbitrary, this holds for all $\pi$.

Now, let's show that $2 \implies 1$. If there exists $X_k$ on every bidirected path between $X_j$ and $X_i$ in $G_{MAG}$, then we know that this $X_k$ must have $\sigma(X_k) > \sigma(X_i), \sigma(X_j)$. Therefore, 1 is satisfied.

$\square$

## A.1 PROOF OF THEOREM 4.1

**Theorem 4.1.** *Take a Bayesian network $G = (V, E)$ and a set of nodes to remove $S \subseteq V$, and let* final_graphs *be the output of running Algorithm 1 on $G, S$. For each $G'$ in* final_graphs, $G'$ *is a Bayesian network, and there exists no conditional independency encoded by $G'$ not encoded by $G$.*

*Proof.* We claim that all of the final graphs are valid. First, since neither the base graph nor any of the component graphs contain any bidirected edges, $G'$ must only contain directed edges. Additionally, it must be a DAG because of the check we perform before returning *final_graphs*. Thus, $G'$ is indeed a Bayesian network.

Next, we wish to show that it has no conditional independencies that $G_{MAG}$ does not have. It suffices to show that if there exists an active path $\pi$ between $A$ and $B$ given $Z$ in $G_{MAG}$, then there exists an active path $\pi'$ between $A$ and $B$ given $Z$ in $G'$; this ensures that if $A \not\perp B \mid Z$ in $G_{MAG}$, then $A \not\perp B \mid Z$ in $G'$.

To find such an active path, it suffices to find a path $\pi'$ that uses a subset of the nodes that $\pi$ used, with the restriction that collider nodes on $\pi'$ were also colliders on $\pi$, and non-collider nodes on $\pi'$ were also non-colliders on $\pi$. In other words, we wish to ensure that the *collider status* of the nodes on $\pi'$ matches the collider status of these nodes on $\pi$.

Consider the active path between $A$ and $B$ given $Z$ of $\pi = (A = X_0, X_1, \ldots, X_n = B)$ in $G_{MAG}$. For $\pi'$, we will take a path that uses a subset of these nodes. For the parts of path $\pi$ that are directed edges, we will take the exact same path in $\pi'$. This is possible because $G'$ contains all of the directed edges that $G_{MAG}$ contains.

For parts of $\pi$ that go through a BCC, say $Y_1 \leftrightarrow Y_2 \leftrightarrow \cdots \leftrightarrow Y_m \subseteq \pi$, we claim that there exists a path in $G'$ that goes through a subset of $(Y_1, \ldots Y_m)$ and ensures that the collider status of these nodes is preserved in $\pi'$.

Before we begin, consider any intermediate node $Y_k$ on our path. We will refer to the "incoming edge for $Y_k$" as the edge between $Y_{k-1}$ and $Y_k$, and the "outgoing edge" is the edge between $Y_k$ and $Y_{k+1}$.

If the incoming edge for $Y_k$ has an arrowhead at $Y_k$, that means we have $Y_{k-1} \xrightarrow{?} Y_k$.

We will do casework on the edges between $(Y_0, Y_1)$ and $(Y_m, Y_{m+1})$, as we wish to preserve if there is an arrowhead or not on the incoming (outgoing) edge for $Y_1$ ($Y_m$) in $\pi'$. Note that if $Y_0$ or $Y_{m+1}$ do not exist ($Y_1 = A$ or $Y_m = B$), then they are an endpoint and do not have a collider status. Thus, we do not care about the arrowhead on the outgoing (incoming) edge for $Y_1$ ($Y_m$), and we can just follow the same steps as case 1 below, but omit the $Y_0$ ($Y_{m+1}$).

**Case 1:** We have $Y_0 \leftarrow Y_1 \leftrightarrow \cdots$ and $\cdots \leftrightarrow Y_m \rightarrow Y_{m+1}$. Then, we have to preserve non-collider status of $Y_1$ and $Y_m$.

As seen in Lemma A.3, our algorithm will add an edge between any two nodes in the same BCC unless there exists some node $V$ on every bidirected path between $Y_1$ and $Y_m$ such that $Y_1 \rightarrow V \leftarrow Y_m$, Hence, we either have

- $Y_1 \leftarrow Y_m$,

- $Y_1 \rightarrow Y_m$, or

- $Y_1 \rightarrow Y_r \leftarrow Y_m$, for some $Y_r$ between $Y_1$ and $Y_m$ in $G'$.

Thus, we can take the path

- $Y_0 \leftarrow Y_1 \leftarrow Y_m \rightarrow Y_{m+1}$,

- $Y_0 \leftarrow Y_1 \rightarrow Y_m \rightarrow Y_{m+1}$, or

- $Y_0 \leftarrow Y_1 \rightarrow Y_r \leftarrow Y_m \rightarrow Y_{m+1}$.

for $\pi'$, respectively. We claim that this will still be an active path. Note that the nodes $Y_1$ and $Y_m$ are still non-colliders (as they were in $\pi$) and the node $Y_r$ is still a collider (as it was in $\pi$). If we initially

had $\leftarrow Y_0 \leftarrow Y_1$ in $G_{MAG}$, then $Y_0$ is still a non-collider in $G'$, and if we initially had $\overset{?}{\to} Y_0 \leftarrow Y_1$ in $G_{MAG}$, then $Y_0$ is still a collider in $G'$. Fundamentally, this is because there is still an arrowhead at $Y_0$ on the outgoing edge of $Y_0$. Similarly, the collider status of $Y_{m+1}$ will be preserved because there is still an arrowhead at $Y_{m+1}$ on the incoming edge of $Y_{m+1}$.

**Case 2:** We have $Y_0 \leftarrow Y_1 \leftrightarrow \cdots$ and $\cdots \leftrightarrow Y_m \leftarrow Y_{m+1}$. Again, we either have

- $Y_1 \leftarrow Y_m$, so we must also have $Y_1 \leftarrow Y_{m+1}$ since $Y_{m+1} \in Pa(Y_m)$,

- $Y_1 \to Y_m$, or

- $Y_1 \to Y_r \leftarrow Y_m$, and $Y_r \leftarrow Y_{m+1}$, for some $Y_r$.

Therefore, in $\pi'$, we can take the paths

- $Y_0 \leftarrow Y_1 \leftarrow Y_{m+1}$,

- $Y_0 \leftarrow Y_1 \to Y_m \leftarrow Y_{m+1}$,

- $Y_0 \leftarrow Y_1 \to Y_r \leftarrow Y_{m+1}$

for $\pi'$. Again, we have that $Y_1$ is still a non-collider, $Y_m$ and $Y_r$ are still a colliders, and there is still an arrowhead at $Y_0$ on its outgoing edge while there is not an arrowhead at $Y_{m+1}$ on its incoming edge.

**Case 3:** We have $Y_0 \to Y_1 \leftrightarrow \cdots$ and $\cdots \leftrightarrow Y_m \to Y_{m+1}$. This is symmetric to the previous case.

**Case 4:** We have $Y_0 \to Y_1 \leftrightarrow \cdots$ and $\cdots \leftrightarrow Y_m \leftarrow Y_{m+1}$. We can have:

- $Y_1 \leftarrow Y_m$, so we must also have $Y_1 \leftarrow Y_{m+1}$,

- $Y_1 \to Y_m$, so we must also have $Y_0 \to Y_m$, or

- $Y_1 \to Y_r \leftarrow Y_m$, as well as $Y_0 \to Y_r \leftarrow Y_{m+1}$, for some $Y_r$.

In $\pi'$, we can take the paths

- $Y_0 \to Y_1 \leftarrow Y_{m+1}$,

- $Y_0 \to Y_m \leftarrow Y_{m+1}$,

- $Y_0 \to Y_r \leftarrow Y_{m+1}$

for $\pi'$. we see that $Y_1, Y_m$, and $Y_r$ are still colliders, while there is no arrowhead at $Y_0$ ($Y_{m+1}$) on its outgoing (incoming) edge.

Therefore, if we go back to our original path $\pi$, for each section of $\pi$ that goes through a BCC, we can take a corresponding path according to the procedure above to get a $\pi'$ that is still active.

Since we are able to show that if there exists an active path between $A$ and $B$ given $Z$ in $G_{MAG}$, then there exists an active path between $i$ and $j$ given $Z$ in $G'$, we know that $G'$ does not contain any conditional independencies that $G_{MAG}$ does not contain, and so $G'$ satisfies the conditions of our theorem.

$\square$

## A.2 PROOF OF THEOREM 4.2

**Theorem 4.2.** *Take a Bayesian network $G = (V, E)$ and a set of nodes to remove $S \subseteq V$, and let* final_graphs *be the output of running Algorithm 1 on $G, S$. For any $G^*$ that satisfies the following conditions, we must have $G^*$ in* final_graphs*:*

- *$G^*$ is a marginalized Bayesian network on the nodes $V \setminus S$.*

- *The conditional independencies encoded by $G^*$ are a subset of the conditional independencies encoded by $G$.*

- *The ancestral relationships established in $G$ are preserved.*

- *$G^*$ contains the minimum number of edges amongst all Bayesian networks satisfying the previous 3 conditions.*

*Proof.* Recall that the base graph $G_{base}$, consists of all of the directed edges in $G_{MAG}$, and that all of the graphs identified and returned by our algorithm are of the form $G_{base} \cup \bigcup_{i=1}^{k} G_{C_i, \sigma_i}$. In order to show that $G^*$ is also of this form, we aim to show that all of these edges are necessary for some choice of $\sigma_i$.

**Lemma A.4.** *$G^*$ must include all of the edges in the base graph.*

*Proof.* Recall that ancestral graphs are closed under marginalization, so the conditional independencies encoded by $G_{MAG}$ are exactly the same as the conditional independencies encoded by $G$ amongst the nodes $V \setminus S$. Therefore, all of the conditional independencies that exist in $G^*$ must also exist in $G_{MAG}$.

Assume for the sake of contradiction that there exists $A, B$ such that $A \rightarrow B$ in $G_{MAG}$ but there exists no edge between $A$ and $B$ in $G^*$. Since $G^*$ is a Bayesian network and $B \notin an(A)$, Lemma A.1 tells us that $A \perp B \mid Pa(B)$. However, this conditional independency does not hold in $G_{MAG}$, as $A \rightarrow B$ is an active path given any conditioning set. Since $A \perp B \mid Pa(B)$ is in $G^*$ but not $G_{MAG}$, we know that $G^*$ is not a valid marginalized graph. This is our desired contradiction. $\square$

Next, since $G^*$ is a Bayesian network, it must be possible to assign a topological ordering $\sigma$ to its nodes. Then, we can obtain an induced topological ordering on the nodes of each BCC in $G_{MAG}$, by taking the topological ordering on the entire graph, and simply deleting the nodes that are not in the BCC.

**Lemma A.5.** *Given any component $C$ and the topological ordering induced by the topological ordering in $G^*$, all of the edges added in Algorithm 2 to $G_C$, a subgraph with nodes $C \cup Pa(C)$, must exist in $G^*$ as well.*

*Proof.* Without loss of generality, given our induced topological ordering, label the nodes in a given BCC as $X_1, \ldots, X_n$, where node $X_i$ has topological order $i$. Consider two arbitrary nodes $X_i$ and $X_j$ with $i \neq j$, and suppose without loss of generality that $i > j$, i.e., that $X_i$ has a higher topological order than $X_j$.

In our algorithm, we add edges $X_j \rightarrow X_i$ and $V \rightarrow X_i$ for $V \in Pa(X_j)$ if and only if there exists a bidirected path $X_j \leftrightarrow X_{j_1} \leftrightarrow \cdots \leftrightarrow X_{j_m} \leftrightarrow X_i$ such that $X_{j_k}$ has lower topological order than $X_i$ for each $k \in \{1, \ldots, m\}$.

Assume for the sake of contradiction that the condition above holds, but we do not have $V \rightarrow X_i$ in $G^*$ for some $V \in \{X_j\} \cup Pa(X_j)$. We aim to show that there exists a conditional independency that is true in $G^*$ but not in $G_{MAG}$ to complete our proof by contradiction. Letting $Pa_{G^*}(X_i)$ denote the parents of $X_i$ in $G^*$, take

$$Z = \begin{cases} Pa_{G^*}(X_i) \cup \{X_{j_1}, \ldots, X_{j_m}\} & \text{if } V = X_j \\ Pa_{G^*}(X_i) \cup \{X_j, X_{j_1}, \ldots, X_{j_m}\} & \text{otherwise} \end{cases}.$$

Since, by hypothesis, the topological ordering of the BCC is induced from that of $G^*$, we know that $X_j, X_{j_1}, \ldots, X_{j_m}$ must also have a lower topological order than $X_i$ in $G^*$, and that $X_i \notin an(V)$. By definition, we also have that $Pa_{G^*}(X_i) \subseteq Z$. Therefore, by Lemma A.1, we have that $X_i \perp V \mid Z$ in $G^*$.

On the other hand, consider the path from $V$ to $X_i$ through $X_j \leftrightarrow X_{j_1} \leftrightarrow \cdots \leftrightarrow X_{j_m} \leftrightarrow X_i$ in $G_{MAG}$. Given our conditioning set $Z$, this path is active, because all of the intermediate nodes are colliders and they are all in $Z \subseteq an(Z)$. Thus, $X_i \not\perp V \mid Z$ in $G_{MAG}$.

This is our desired contradiction, as we cannot have a conditional independency that exists in $G^*$ but not $G_{MAG}$. Thus, we must actually have $V \to X_i$ in $G^*$, and since $i, j$ were arbitrary, every edge added by our algorithm to $G_C$ must also exist in $G^*$. $\qquad\square$

Let's put together Lemma A.4 and Lemma A.5. These tell us that all of the edges in our base graph are in $G^*$, and if we run our algorithm on the topological order induced by $G^*$ for each component, we will not add any edges to each $G_C$ that do not exist in $G^*$. Since $G^*$ is minimal, we must have that $G^*$ is equal to $G_{base} \cup \bigcup_{i=1}^{k} G_{C_i}$. Since our algorithm iterates through all possible topological orderings for each BCC, it will necessarily test the orderings that are induced by $G^*$, and thus identifies $G^* = G_{base} \cup \bigcup_{i=1}^{k} G_{C_i}$ as a valid marginalized Bayesian network.

It remains to show that our algorithm correctly adds $G^*$ to its *final_edges*. After getting the resulting $G_C$ graphs for each of the components, the algorithm proposes final graphs that minimize the sum of the number of edges added to each component. We claim that this is an equivalent heuristic to minimizing the number of edges in the final graph.

The number of edges in the final graph is the number of edges in $G_{base} \cup \bigcup_{i=1}^{k} G_{C_i}$ amongst all choices of $G_C$. This is equivalent to minimizing the number of edges in $\bigcup_{i=1}^{k} G_{C_i}$, as recall that we only add edges to component graphs if they do not already exist in the base graph. Thus, it suffices to show that no edge is added to two different $G_{C_i}$ component graphs, as then we will have

$$\left| \bigcup_{i=1}^{k} G_{C_i} \right| = \sum_{i=1}^{k} |G_{C_i}|.$$

When we add edges to $G_C$, we only add edges that have arrowheads at some nodes in the bidirected component itself (although edges may originate from nodes in the bidirected component or its parents). Each node is part of at most one bidirected component, as if there exists a bidirected path from $A$ to $B$ and $A$ to $C$, then there also exists a bidirected path from $A$ to $C$. Therefore, since the nodes in the bidirected components are disjoint, the edges added to different components must be disjoint as well.

In conclusion, this implies that it suffices to minimize the sum of the number of edges in each component. Therefore, our algorithm is able to identify $G^* = G_{base} \cup \bigcup_{i=1}^{k} G_{C_i}$ as a graph with minimal edges, and it is added to *final_graphs*.

$\qquad\square$

### A.3 SUB-OPTIMALITY OF GREEDILY REVERSING EDGES

As an illustrative example, consider the graph in Figure 3. Reversing $R \to A$ would add 6 edges ($\{D, E, F\}$ to $\{B, C\}$), while reversing $R \to B$ would only add 5 edges ($G \to C$ and $\{G, H, I, J\}$ to $A$). If we greedily chose $R \to B$, we would either add 3 more edges (if we then reversed $R \to A$)

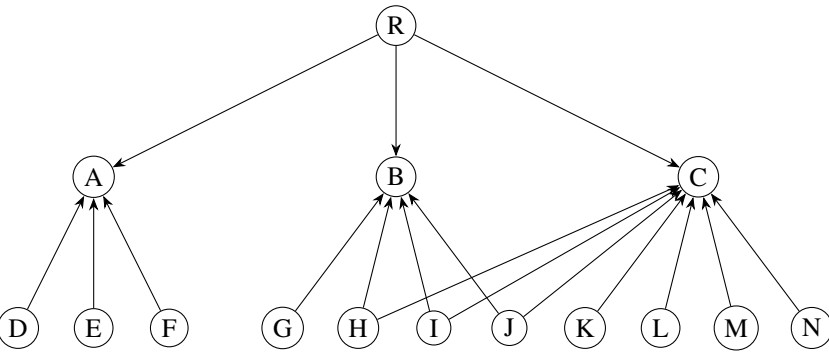

Figure 3: An example of a Bayesian network where greedily choosing to reverse edges when marginalizing out node $R$ is not optimal.

or 4 more edges (if we then reversed $R \to C$), for a total of at least 8 edges. On the other hand, if we reversed $R \to A$ first initially, we would only have to add 1 more edge when reversing $R \to B$ (the edge $G \to C$), for a total of 7 edges.

### A.4 THE USE OF LARGE LANGUAGE MODELS

Large language models were used to help us more easily find the previous work that had been completed on marginalization in Bayesian networks, and related concepts, such as acyclic directed mixed graphs and maximal ancestral graphs. Additionally, they were used to help create the Bayesian networks displayed in figures 1 and 3, and to debug code.

