# OpenReview forum: "An Optimal Algorithm for Marginalization in Bayesian Networks"
_ICLR.cc/2026/Conference — ICLR 2026 Conference Withdrawn Submission_

### Official Review · Reviewer_DHoK · 2025-10-26

**Soundness:** 2
**Presentation:** 1
**Contribution:** 1
**Rating:** 2
**Confidence:** 3

**Summary:**

The submission studies marginalization in the context of Bayesian Networks (BNs). In particular, it provides a procedure to compute a BN that omits a specified set of nodes while preserving independencies and ancestor/descendant relations.

**Strengths:**

Bayesian Networks are a well-established model that is of interest to both empirical and foundational subcommunities in ICLR.

**Weaknesses:**

Given that the submission focuses on marginalization in BNs, the discussion of related work in that context is lackluster. The concept of marginalization in BNs is introduced as if it were a well-established concept, but in the relevant 4th paragraph of the introduction there is only a single reference given to previous works considering marginalization in BNs - and that reference is barely relevant (it studies abstractions of BNs and only contains a single occurrence of the word "marginalization", as a by-the-way remark). If marginalization in BNs has been understudied, the introduction should have made a point of that. It seems there are several related works on marginalization in adjacent graphical models (such as Markov models) from the early 2000s as discussed in Section 2.2, and perhaps some of these can be transferred to the BN setting - however, a discussion of whether this is the case is missing.

Moreover, the definitions are not provided with sufficient formal rigor, leading to ambiguities which would otherwise be easy to avoid. For example, ancestral graphs are introduced as the extension of BNs via the inclusion of bidirectional edges, yet how these new bidirectional edges interact with the previously introduced notation for purely one-directional edges is unclear. In classical graph-theoretic terms, bidirectional edges would be interpreted as two directed arcs which form a C2 - yet that is clearly not the case here, given the claim that ancestral graphs must be acyclic. But then how do the ancestor and descendant relations interact with bidirectional edges? Can paths contain bidirectional edges?

Another issue is that the specific contributions are not clearly described. As one example of this, the problem solved by Algorithm 1 (formalized only in Section 4) asks a set of BNs instead of a single BN and this is neither explained nor discussed. Has this specific problem been studied before, or how closely is it related to the previous work on BNs? As another example, there is no discussion of the running time of the proposed algorithm - without checking the algorithm as well as the referenced subprocedures, one cannot even ascertain whether it runs in polynomial time. And the submission is also missing a description or discussion of the technical contributions underlying the foundational results: what were the main difficulties that had to be overcome (or novel insights that had to be obtained) towards establishing the new results?

There are also other, smaller issues such as:

-The very first paragraph of the introduction is repeated (probably just a copy-paste error).

-At the end of Section 3.2, the submission claims that Shachter's reduction operations need not produce the most compact BN even after considering all possible orders of edge reversals and node removals. Since this claim is crucial for the significance of the submission, I would have expected this claim to be substantiated by a reference or explanation.

-There is still ample space left in the main body of the submission, even though several important proof details have been delegated to the appendix without being reflected in the short proof sketches. One wonders why has the remaining space not been used in any way.

**Questions:**

The authors are welcome to respond to any of the concerns or questions raised above.

---

### Official Review · Reviewer_65Du · 2025-10-27

**Soundness:** 3
**Presentation:** 3
**Contribution:** 3
**Rating:** 8
**Confidence:** 2

**Summary:**

The paper tackles the problem of graph marginalization in Bayesian networks and provides an algorithm for finding the optimal marginalization. The definition of a valid marginalization and optimality is given. The algorithm leverages the closed-ness of MAGs (maximal ancestral graphs) under marginalization, and decomposes the MAG into bidirected connected components (BCCs) and enumerates valid topological orders. The algorithm is shown to be correct and sound. Emprical experiments are conducted to show consistent speedup against existing algorithms.

**Strengths:**

- This is the first graph marginalization algorithm with a provable optimality guarantee. Clear theoretical advancement.
- The experiments demonstrates strong speedups in a wide set of random graph configurations.
- This paper is generally well-written. The introduction and motivation are clear.

**Weaknesses:**

- It would be better to have a running example (such a Figure 1 in the manuscript) to illustrate (i) how the proposed algorithm works; and (ii) why the existing algorithms fail to give valid or optimal marginalization. Maybe the example in Figure 1 is not complex enough.

**Questions:**

- The experiment only demonstrates the speedup compared to the existing algorithms rather than the correctness of the outputs. I wonder empirically for the graphs considered in the experiments, why is the percentage that the existing algorithms fail to give a valid or optimal marginalization?
- The performance of the algorithm appears to be driven by BCC structure. How large can BCCs typically get in practice?
- Is there a runtime complexity analysis for the proposed algorithm?

---

### Official Review · Reviewer_mPk8 · 2025-10-29

**Soundness:** 3
**Presentation:** 3
**Contribution:** 1
**Rating:** 2
**Confidence:** 5

**Summary:**

The paper studies the problem, given a Bayesian network over a set of variables, to find the smallest Bayesian network for a given subset of variables that introduces no extra conditional independencies and does not violate the ancestral constraints of the original Bayesian network. It proposes the first provably optimal algorithm for this problem, i.e., one that guarantees to find a Bayesian network with minimal number of edges.

**Strengths:**

The paper is generally well-written. I had no problems of following the arguments laid out in the paper and the proposed algorithm appears to be sound. Moreover, the algorithm is novel and the first to (fully) address this problem.

**Weaknesses:**

The main weaknesses of the paper are that the problem is rather niche/specialized and that, while the proposed algorithm is sound, no formal run-time analysis is provided (the algorithm runs in exponential time) nor a complexity theoretic classification of the problem (e.g. NP-hardness proof).

Niche problem:
The problem that is solved is effectively, for a given maximal ancestral graph (MAG), to find the smallest Bayesian network with no extra condititional independencies and satisfying given ancestral constraints (given by a partial order). This can be used to tackle the (maybe more natural) problem of reducing/marginalizing a Bayesian network to a subset of variables while preserving its ancestral constraints, introducing no further conditional independencies and adding as few edges as possible. However, it remains unclear how solving this task can be useful in a larger pipeline. Also for causal inference tasks, the MAG representation appears to be more useful than the Bayesian network that is constructed from it in this procedure. This raises the question why it is crucial to keep the ancestral constraints, which is motivated due to preserving the causal ordering.

Formal complexity analysis:
The provided algorithm has exponential run-time. While it is certainly much faster than naive exponential-time algorithms, an analysis corroborating this and discussing the worst-case run-time is needed. Also, from the paper it is not clear to me whether the hardness of the given problem has been studied. Even if this is simple/obvious, a proof/discussion of e.g. it's NP-hardness would complement the exponential run-time algorithm.

**Questions:**

1. Can you give examples how the studied problem can be useful for larger pipelines? Are there (naive) implementations in software libraries (Bayesian networks, causal inference) of algorithms tackling this or similar problems?

2. Is it possible to say whether the problem is (likely) NP-hard? Is there hope that a polynomial-time algorithm can be derived?

Suggestions/typos:
- line 062: "the topological ordering", rephrase to not sound like there is a unique topological ordering
- line 169: removing barren nodes, i.e. nodes that...
- line 206: return a *non-empty* set of BNs ...
- line 250: "independency" should be independence
- line 262: "we'll" replace by "we will" (same for "let's" in line 357)
- line 367: "in our algorithm of this method", do you mean "in our implementation of this method"?

---

### Official Review · Reviewer_9W9b · 2025-10-31

**Soundness:** 3
**Presentation:** 3
**Contribution:** 2
**Rating:** 2
**Confidence:** 4

**Summary:**

The paper contributes with a new algorithm to obtain a marginalized Bayesian Network, that is, a Bayesian Network DAG approximating the projection of an input Bayesian Network DAG under the following constraints: it must preserver the ancestral relations (or topological constraints) of the original DAG, it must preserve the dependences of the original DAG, and it must minimize the number of edges. The authors claim this to be the first algorithm able to ensure that a edge-minimal network is found. The algorithm performs a local brute-force over the bi-directed components of a Maximum Ancestral graph representation of the input DAG in order to find a candidate DAG, by enumerating local topological orderings. The authors prove that this approach ensures sufficiency and necessity, proving soundness of the algorithm. Experiments with synthetic networks show a significant speedup compared to a naive approach.

**Strengths:**

- Text is relatively easy to follow (for someone with proper background)
- Sound approach
- Unexplored task

**Weaknesses:**

- Lack of proper motivation for the task
- Brute-force approach scales poorly (dismissing the single motivation given in the introduction, to be able to speed up inference) in the size of biconnected components
- Experiments are very preliminary and use unrealistic network structures

**Questions:**

The work misses a proper justification. Bayesian networks with thousands of nodes are not very interpretable; in fact, a neural network that uses activation functions with image in [0,1] can be seen as a Bayesian network over binary variables, which shows that this fact alone does not contribute for interpretability. As it is, it is not clear that marginalization of Bayesian Networks can help in probabilistic inference; it is possible that the task is NP-hard in itself and there is no evidence that inferences produced by a marginal Bayesian network are good approximations (as the graph might have high in-degree and conditional probability estimates are less statistically robust). In fact, one way of obtaining the parameters of the marginal Bayesian network is by doing probabilistic inference in the original network.

The description of the task is very poorly motivated. Why one would like to obtain a marginal Bayesian network that respects the ancestral relations? With respect to causal modeling, one can often work with the ADMG (which admits a factorization), thus dispensing with the approximations inserted by marginal DAG. If one is only interested in a probabilistic model, than keeping the ancestral relations seem unnecessary (and can lead to much larger models).

Lemma A.1 (and Cor A.1.1.) are well-known, see e.g. Koller and Friedman 2009.

---

### Note · Authors · 2025-11-26

**Comment:**

Thank you to the reviewers for taking the time to read our paper and give detailed feedback! We plan to resubmit the work with clearer motivation, additional runtime analysis, and more comprehensive experiments.

**Withdrawal Confirmation:**

I have read and agree with the venue's withdrawal policy on behalf of myself and my co-authors.